DNA methylation and gene expression profiling reveal potential association of retinol metabolism related genes with hepatocellular carcinoma development

Zhao Yanteng zyt198910066@126.com 1
Wan Kangkang 2
Wang Jing 1
Wang Shuya 1
Chang Yanli 1
Du Zhuanyun 1
Zhang Lianglu 2
Dong Lanlan 2
Zhou Dihan 2
Zhang Wei 2
Wang Shaochi 3
Yang Qiankun yqiankun@126.com 1
1 Department of Transfusion, The First Affiliated Hospital of Zhengzhou University , Zhengzhou , Henan Province , China
2 Wuhan Ammunition Life-tech Company, Ltd., CN , Wuhan , Hubei Province , China
3 Center for Translational Medicine, The First Affiliated Hospital of Zhengzhou University , Zhengzhou , China
Pfeffer Ulrich
Electronic publication date: 2024 Aug 23
Publication date: 2024
Volume: 12
Electronic Location ID: e17916
Received 2024 Mar 21; Accepted 2024 Jul 23
Copyright: ©2024 Zhao et al.
Copyright year: 2024
Copyright holder: Zhao et al.
License: This is an open access article distributed under the terms of the Creative Commons Attribution License, which permits unrestricted use, distribution, reproduction and adaptation in any medium and for any purpose provided that it is properly attributed. For attribution, the original author(s), title, publication source (PeerJ) and either DOI or URL of the article must be cited.
License URL: https://creativecommons.org/licenses/by/4.0/

Keywords: Hepatocellular carcinoma, DNA methylation, Gene expression, Retinol metabolic related genes

Funding: Henan Province 232102310028 This study was supported by science and technology project of Henan Province (232102310028). The funders had no role in study design, data collection and analysis, decision to publish, or preparation of the manuscript.

==============================
Background

Aberrant DNA methylation patterns play a critical role in the development of hepatocellular carcinoma (HCC). However, the molecular mechanisms associated with these aberrantly methylated genes remain unclear. This study aimed to comprehensively investigate the methylation-driven gene expression alterations in HCC using a multi-omics dataset.

Methods

Whole genome bisulfite sequencing (WGBS) and RNA sequencing (RNA-seq) techniques were used to assess the methylation and gene expression profiles of HCC tissues (HCCs) and normal adjacent tissues (NATs). The candidate genes’ potential function was further investigated using single-cell RNA sequencing (scRNA seq) data.

Results

We observed widespread hypomethylation in HCCs compared to NATs. Methylation levels in distinct genomic regions exhibited significant differences between HCCs and NATs. We identified 247,632 differentially methylated regions (DMRs) and 4,926 differentially expressed genes (DEGs) between HCCs and NATs. Integrated analysis of DNA methylation and RNA-seq data identified 987 methylation-driven candidate genes, with 970 showing upregulation and 17 showing downregulation. Four genes involved in the retinol metabolic pathway, namely ADH1A, CYP2A6, CYP2C8, and CYP2C19, were identified as hyper-downregulated genes. Their expression levels could stratify HCCs into three subgroups with distinct survival outcomes, immune cell infiltration, and tumor microenvironments. Validation of these findings in an independent dataset yielded similar outcomes, confirming the high concordance and potential prognostic value of these genes. ScRNA seq data revealed the low expression of these genes in immune cells, emphasizing their role in promoting malignant cell proliferation and migration. In conclusion, this study provides insights into the molecular characteristics of HCC, revealing the involvement of retinol metabolism-related genes in the development and progression of HCC. These findings have implications for HCC diagnosis, prognosis prediction, and the development of therapeutic targets.

Introduction

Hepatocellular carcinoma (HCC) is a primary malignant tumor of the liver that predominantly occurs in patients with chronic liver disease and cirrhosis. Over the past two decades, the incidence of HCC has been on the rise globally (Sung et al., 2021), reaching the ranks of the five leading causes of cancer-related disability-adjusted life years by 2019 (Kocarnik et al., 2022). The incidence of HCC is highest in Asia and Africa, primarily due to the high prevalence of hepatitis B/C virus infection. However, our understanding of the molecular epigenetic events during HCC malignant transformation remains limited.

DNA methylation is a common epigenetic modification. Aberrant methylation patterns are closely associated with the onset of numerous diseases, including HCC (Schübeler, 2015; Nishiyama & Nakanishi, 2021). Previous studies have indicated that advanced HCC exhibits elevated methylation of CHFR and SYK, potential tumor suppressors, while demonstrating decreased methylation levels in LINE-1 and satellite 2 repeat elements. These elements are associated with the progression of chronic hepatitis and cirrhosis to HCC (Hattori & Ushijima, 2016). The application of high-throughput technology has led to the identification of several abnormal methylation events in genomic DNA, which can be used to predict an increased risk of HCC in individuals (Gonçalves et al., 2022; Bai et al., 2021; Li et al., 2020). In addition, a subgroup designated the CpG island methylator phenotype (CIMP) has been identified in HCC based on the Infinium Human Methylation 450k BeadChip data, exhibiting a poorer prognosis than the non-CIMP group (Cheng et al., 2018). These findings suggest that DNA methylation plays a crucial role in the initiation and progression of HCC.

However, the molecular mechanisms associated with aberrantly methylated genes are rarely reported. Unlike single nucleotide variants (SNVs) that alter the structure of a gene, DNA methylation that occurs in the regulatory region of a target gene is usually thought to result in gene silencing or down-regulation of expression. The integration of DNA methylation and RNA expression data offers a valuable opportunity to elucidate the molecular mechanisms underlying aberrantly methylated genes. Udali et al. (2015) identified a number of epigenetically regulated candidate tumor-suppressor genes, such as the retinol metabolism genes, in alcohol-associated hepatocarcinogenesis. However, the relationship between clinical features and prognosis of HCC patients and the low expression of retinol metabolism genes is unclear. Huang et al. (2021) identified 611 differentially methylated region-associated, differentially expressed genes (DMR-DEGs) by comparing DNA methylation and expression in tumors and adjacent tissues from 33 HCC patients using WGBS and RNA-seq. They depicted activated pathways, such as those for the cell cycle and DNA repair, as well as repressed key metabolic pathways, such as tyrosine and monocarboxylic acid metabolism, induced by aberrant DNA methylation of promoters and enhancers in HCC. It is necessary to clarify whether other critical pathways and methylation-driven hub genes exist during HCC development, and the relationship of methylation-driven hub genes with mutation and the tumor microenvironment remains unknown and needs to be validated.

Understanding the molecular pathogenesis of HCC is important for the development of effective diagnostic and therapeutic approaches, as the disease is diverse and complex, involving various underlying molecular pathways. Given these considerations, this study was designed to investigate the methylation-driven deregulated expression of genes in HCC using a large cohort of patients with diverse clinical features. Meanwhile, the biological pathways that may be influenced by these genes were also explored. Our findings provide an overview of the molecular characteristics of HCC, with implications for the diagnosis, prognosis prediction, and treatments of HCC.

Materials & Methods

Samples collection

A total of 12 HCC patients from the First Affiliated Hospital of Zhengzhou University were recruited for this study. Fresh tumor tissue and paired NAT were collected from each patient. All samples were immediately frozen and preserved in liquid nitrogen following surgical resection. The clinical characteristics of the 12 patients are shown in Table S1. All participants signed the informed consent forms and were informed of the purpose for which the samples would be used. The Ethics Committee of the First Affiliated Hospital of Zhengzhou University granted approval for this study (number 2022-KY-0631-002) in accordance with the ethical guidelines outlined in the 1964 Declaration of Helsinki.

Data preparation

The public data for the PRJNA762641 project was derived from a study conducted by Huang et al. (2021), which includes WGBS and RNA-seq data. The raw data for the 33 HCC samples and their paired NATs were obtained from the Sequence Read Archive (SRA) website (https://www.ncbi.nlm.nih.gov/bioproject/?term=PRJNA762641). The information pertinent to 33 HCC patients is listed in Table S2. The GSE70090 dataset was derived from the study conducted by Li et al. (2016), which included WGBS data for 28 normal and lung/liver cancer tissues. In this study, only the raw read files of four HCCs and their matched NATs in the GSE70090 dataset were utilized (Table S3).

In addition, we obtained HCC data from The Cancer Genome Atlas (TCGA) database (https://portal.gdc.cancer.gov/), which encompassed patient clinical information, 450k methylation microarray data, RNA-seq data, and genomic variation. The RNA-seq data for the ICGC LIHC dataset was downloaded from the ICGC Data Portal (https://dcc.icgc.org/). The details of all datasets used in the study are listed in Table S4.

Analysis of single-cell RNA sequencing (scRNA seq) data

The GSE151530 (Ma et al., 2021) dataset comprises scRNA seq data of 46 samples from 37 patients with liver disease. Of these patients, 25 had HCC and 12 had cholangiocarcinoma (CC). The samples from CC patients were removed, and the remaining 50,023 cells were obtained. The following analyses were performed using Seurat (version 4.1.0) (Hao et al., 2021). Initially, samples with fewer than 200 cells were excluded. Additionally, cells with a mitochondrial RNA content exceeding 10% were excluded from the analysis. Totally, 49,094 cells that met the abovementioned criteria were obtained from 23 samples collected from 17 HCC patients. The Seurat integration workflow was employed to integrate single-cell data across different samples with the default parameters. Scaled z-scores for each gene in the integrated data were calculated using the ScaleData function, which was then used as input for principal component analysis (PCA). The unsupervised clustering of cells was conducted using the FindClusters function. Only the top 2,000 most variable genes were utilized, with the resolution parameter set to 0.5. The retinol score for each cell was determined by the average expression of three genes: CYP2C8, CYP2A6, and ADH1A, and each gene was given the same weight (Knudsen et al., 2014).

WGBS and methylation calling

Genomic DNA was extracted from fresh tissue samples (Qiagen, Germany) and treated with bisulfite using the EZ DNA Methylation Gold Kit (Zymo Research, Tustin, CA, USA), followed by purification to obtain the bisulfite-treated DNA. The bisulfite-treated DNA was then randomly interrupted and ligated with adapters to construct sequencing libraries. Subsequently, whole genome sequencing was performed on the libraries using the high-throughput sequencing platform DNBSEQ. Next, the raw reads were filtered using the SOAPnuke tool (Chen et al., 2018) to eliminate low-quality reads, adapter contamination, and other artifacts. The parameters used for filtering were as follows: ‘-n 0.001 -l 20 -q 0.4--adaMR 0.25--ada_trim--polyX 50’. Finally, the clean reads were obtained and their base quality was set to Phred+33.

The initial step in analyzing WGBS data involves aligning the short sequence reads to the reference genome, followed by quantifying the C-to-T conversions in the aligned reads at each CpG site. In this study, the Bismark tool (Krueger & Andrews, 2011) was utilized for methylation calling. The reference genome sequence (GRCh38) was initially indexed using the Bismark genome preparation tool, bismark_genome_preparation. Then, the clean reads were aligned to the reference genome using bowtie2. The Bismark methylation extractor tool was employed to extract methylation values for each CpG. Methylation values were calculated as the percentage of methylated cytosines at each CpG site across all reads covering that site.

Differential methylation analysis

Three datasets, PRJNA762641, GSE70090, and our customized data (PRJNA984754), were integrated into a single dataset for differential methylation analysis. One NAT sample (SRR2074679) from GSE70090 was excluded from further analysis due to the unavailability of the complete fastq file. Consequently, WGBS data were acquired from 97 samples, comprising 48 NATs and 49 HCCs. The differentially methylated CpGs (DMCs) were identified by comparing the methylation levels between NATs and HCCs using a rank-sum test followed by multiple testing corrections. A significant DMC was defined as a result of false discovery rate (FDR) being less than 0.05. Abnormally methylated regions were identified using a circular binary segmentation algorithm (CBS) (Olshen et al., 2004), which has been previously reported for the identification of differentially methylated regions (DMRs) (Gong & Purdom, 2020; Jühling et al., 2016). In brief, the average methylation value for each CpG site was calculated separately for HCC and NAT samples, The delta (Δvalue) value of each CpG was calculated using the formula: Taverage–Naverage, where Taverage and Naverage represent the average methylation value of the CpG in HCC and NAT samples, respectively. DMRs were determined using the CBS algorithm, with a maximum size of less than 1 Mb and at least three CpGs covered.

DMRs were categorized as hypermethylated DMRs (hyper-DMRs) and hypomethylated DMRs (hypo-DMRs), based on the direction of methylation events. Hyper-DMRs were defined as those with a Δvalue ≥ 10, while hypo-DMRs were defined as those with a Δvalue ≤  − 15. The two thresholds were selected for the following reason. The distribution of the Δvalues of DMRs exhibited multimodal distribution patterns, with three peaks around −15, 0, and 10, respectively. The threshold of 10 represents the majority of DMRs that are hypermethylated in HCCs but hypomethylated in NATs. The threshold of −15 represents the majority of DMRs that are hypermethylated in NATs but hypomethylated in HCCs. The locations of DMRs were annotated according to the genomic annotation information (GRCh38) as follows: Upstream (within 2000 bp of the gene TSS), Upstream-Body, Inner-genic, Body, Body-Downstream, Downstream (within 200 bp downstream of the gene), and Intergenic.

The association of DMRs with genes is based on the relative position of DMRs to genes. Specifically, if a specific DMR overlaps (overlap proportion ≥ 50% of the DMR length) or is located within the upstream regulatory region (promoter) of a gene, this DMR is considered to be associated with the upstream regulatory region (promoter) of that gene. Similarly, if a DMR overlaps (overlap proportion ≥ 50% of the DMR length) or is located within the downstream regulatory region, it is deemed to be associated with the downstream regulatory region.

Analysis of CpG genomic distribution

The genome annotation information was obtained from UCSC (https://hgdownload.soe.ucsc.edu/goldenPath/hg38/bigZips/genes/hg38.refGene.gtf.gz). Given that a gene is typically associated with multiple transcripts, we only retained the longest transcript in order to avoid the gene being counted multiple times. The genome sequence was divided into six regions based on the transcript locations: 5′ UTR upstream 5k, 5′ UTR upstream 2k, 5′ UTR, Transcript, 3′ UTR, and 3′ UTR downstream 2kb. The genomic regulatory element information was obtained from the annotation files provided by the NCBI database (https://www.ncbi.nlm.nih.gov/datasets/genome/GCF_000001405.40/). The regulatory elements include Enhancer, Promoter, Silencer, Cis-regulatory region, and Insulator. To evaluate the methylation levels in these regions, we calculated the average methylation values of all CpG sites in HCCs and NATs, respectively.

Differential expression analysis

The RNA-seq data of tumor tissues and matched NATs from 33 HCC patients in the PRJNA762641 project were employed to identify differentially expressed genes (DEGs) between normal and tumor samples. Transcripts per million (TPM) values for all genes in both HCCs and NATs were calculated using RSEM (RNA-Seq by Expectation-Maximization). The rank-sum test was used to identify significant DEGs (FDR < 0.05). Low expressed genes with an average TPM < 10 across all samples were excluded. Methylation-driven DEGs (Methy-DEGs) were those hyper-DMGs with low expression levels or hypo-DMGs with high expression levels in HCCs. The identified DEGs were further analyzed using gene ontology (GO) and pathway enrichment analyses to determine the biological processes and molecular pathways disturbed in HCCs. It should be noted that hyper-DMGs with high expression and hypo-DMGs with low expression in HCCs were not within the scope of this study.

Functional enrichment analysis

We used the ‘clusterProfiler’ tool (Wu et al., 2021) to perform functional enrichment analysis on the DEGs to reveal their potential biological functions. The Gene Ontology database (http://geneontology.org) was used to identify biological processes (BP), cellular components (CC) and molecular functions (MF). Statistical significance was evaluated by multiple tests of the p-values of enriched functions with Bonferroni correction. If the adjusted p-value is less than 0.05, the GO term is considered to be significantly enriched.

Statistical analysis

Both data processing and analysis are implemented in R software (version 4.1.1). We used the Kaplan–Meier method to estimate the survival curves for multiple groups and performed the log-rank tests to assess the differences. Samples with less than 30 days of follow-up or missing survival information were excluded. Mutation enrichment analysis was performed using the ‘clinicalEnrichment’ function within the ‘maftools’ package (Mayakonda et al., 2018), which can perform various groupwise and pairwise comparisons to identify enriched mutations in each category of clinical features. ROC curve analysis was performed using the R package ‘pROC’, and the area under ROC curve (AUC) value was calculated to evaluate the performance of the model. We used the ESTIMATE tool (https://bioinformatics.mdanderson.org/estimate/index.html) to calculate the immune score, stromal score, and ESTIMATE score for the three HCC subgroups. The continuous variables were compared using rank-sum test or Kruskal test, depending on the number of groups being compared and the distribution of data. The categorical variables were compared using Chi-square test or Fisher’s exact test, depending on the expected cell counts. A p-value or FDR < 0.05 was considered statistically significant for all analyses.

Results

Genome-wide methylation characteristics of HCCs and NATs

Following the removal of one outlier (16A), the quality control results showed that the remaining 23 samples in the customized dataset exhibited high-quality WGBS data, with an average sequencing depth of 21×(range 16×–26×), base quality of 31, and mapped quality of 34 (Table S5). The customized WGBS data were integrated with previous public datasets PRJNA762641 and GSE70090. The methylation levels for identical CpG sites showed high concordance across the three datasets (Fig. S1, correlation coefficient > 0.85). The multidimensional scaling plot demonstrated a clear distinction between tumor and normal samples (Fig. 1A), suggesting minimal variability between datasets. The tumor tissues exhibited widespread hypomethylation in comparison to the normal samples (Fig. 1B). Comparing of methylation values between each tumor and its corresponding NAT samples in the customized dataset verified the overall hypomethylation in HCC, as shown in Fig. S2.

Figure 1 Genome-wide methylation characteristics of HCCs and NATs.

(A) The MDS plot shows the distribution of HCCs and NATs in the integrated dataset. (B) Density curves demonstrating the distribution patterns of CpG methylation levels between HCCs and NATs. (C) Methylation levels in different genomic regions between HCCs and NATs. (D) Methylation levels in different regulatory elements between HCCs and NATs.

The methylation levels in different genomic regions of HCCs were found to be decreased compared to NATs, as showed in Fig. 1C. The methylation levels in the regulatory regions (5′ UTR and 5′ UTR upstream 2kb) were lower than that in other regions in both NATs and HCCs. The 5′ UTR upstream 2kb region displayed the lowest level of methylation in NATs. In contrast, the 5′ UTR region located 2kb upstream and the 5′ UTR region of HCC showed the lowest methylation levels in HCCs (Fig. 1C). These results suggest that demethylation events occurring in gene regulatory regions, which are crucial for regulating gene expression, may be of greater significance. In addition, the methylation levels of many regulatory elements were significantly lower in HCCs compared to NATs (Fig. 1D). The enhancer exhibits the lowest methylation level in HCCs, whereas the promoter demonstrates the lowest methylation level in NATs, suggesting a potential methylation bias among different regulatory elements. These results reveal an intricate methylation landscape that enables the distinction between HCCs and NATs.

Differential methylation analysis based on WGBS data

The circular binary segmentation algorithm was employed to identify 247,632 DMRs between HCCs and NATs. Of these, 12,317 were hypermethylated and 120,456 were hypomethylated in HCCs (Fig. 2A). The remaining 114,859 DMRs were classified as “others” due to their methylation values falling between −15 and 10. The hypermethylated DMRs (Hyper-DMRs) exhibited a significantly shorter length than the hypomethylated (Hypo-DMRs) and the “others” of DMRs (Fig. 2B), and had the lowest number of CpGs (Fig. 2C). In addition, the average distance between neighboring CpG sites in Hyper-DMRs was smaller than Hypo-DMRs and the “others” (Fig. 2D). These findings reveal that hypermethylation tends to cluster in compact regions, such as CpG islands.

Figure 2 Differential methylation analysis between HCCs and NATs.

(A) Pie chart shows the proportions of hyper-DMR, hypo-DMR and other types of DMR. (B) Box-plot shows the length distributions of hyper-DMR, hypo-DMR and other types of DMR. (C) Box-plot shows the CpG numbers of hyper-DMR, hypo-DMR and other types of DMR. (D) Box-plot shows the average distance of two adjacent CpGs in hyper-DMR, hypo-DMR and other types of DMR. (E) Distribution of hyper-DMR, hypo-DMR and other types of DMR across different genomic locations.

Regarding the distribution of DMRs, more than 75% of Hyper-DMRs and approximately 45% of Hypo-DMRs were located within genes. A total of 10.8% of Hyper-DMRs and 33.83% of Hypo-DMRs were located in the intergenic regions (Fig. 2E). The total proportion of Hyper-DMRs in the Upstream and Upstream-Body regions was 8.67%, while it was 7.33% for Hypo-DMRs in these two regions. In the Body-Downstream and Downstream regions, the proportions of Hyper-DMRs and Hypo-DMRs were 4.23% and 8.61%, respectively (Fig. 2E). Additionally, 5.54% of Hypo-DMRs spanned the entire gene (Whole-body), whereas, only 0.37% of Hyper-DMRs covered the gene body (Fig. 2E).

The top 100 DMRs with the most significant differences are listed in Table S6. The unbiased genome-wide analysis revealed widespread differential methylation between HCCs and NATs.

Integration of DNA methylation and gene expression data

Integrated analysis of DNA methylation and gene expression data from dataset PRJNA762641 revealed extensive methylation-driven transcriptional deregulation. A total of 4,926 DEGs were identified in HCCs, with 4,662 upregulated and 264 downregulated genes. The prevalence of upregulation (48.67% vs. 2.75%) suggests extensive transcriptional activation events during HCC development (Fig. 3A). This finding is consistent with the frequent occurrence of hypomethylation events in tumors. The functional enrichment analysis showed that the upregulated DEGs in HCC were mainly enriched in biological processes related to RNA metabolism, such as RNA splicing and ribonucleoprotein complex biogenesis (Fig. 3B). In contrast, the downregulated DEGs were significantly enriched in distinct biological processes, mainly in small molecule and organic acid metabolism (Fig. 3C).

Figure 3 Differential expression analysis between HCCs and NATs.

(A) Proportions of upregulated and downregulated DEGs. (B) The significantly enriched GO terms of upregulated DEGs (top 20). (C) The significantly enriched GO terms of downregulated DEGs (top 20). (D) Comparisons of the four categories of DEGs. (E) The significantly enriched GO terms of hypo-upregulated genes. (F) The significantly enriched GO terms of hyper-downregulated genes.

To identify genes with expression changes linked to differential methylation, we associated DMRs with DEGs based on their relative positions. Then, DEGs were classified as hyper-upregulated, hyper-downregulated, hypo-upregulated, or hypo-downregulated according to the direction of methylation and expression changes. We focused on genes that were hyper-downregulated or hypo-upregulated, based on the hypothesis that methylation is negatively correlated with expression, as mentioned in the methods. Consequently, 987 methylation-driven candidate genes were identified, comprising 970 upregulated genes and 17 downregulated genes (Fig. 3D). The 17 hypermethylated and downregulated genes were listed in Table S7. The methylation-associated DEGs account for 20% of the total DEGs, highlighting the significant influence of DNA methylation on gene expression in HCC.

Further analysis revealed that the hypo-upregulated genes were enriched for RNA biosynthesis and transport (Fig. 3E), while the hyper-downregulated genes were associated with retinol, hormone, and olefinic acid metabolism (Fig. 3F). Notably, 4 of the 17 hyper-downregulated genes (ADH1A, CYP2A6, CYP2C8, CYP2C19) are involved in retinol metabolism pathways. Correlation analysis further confirmed that methylation levels of CpGs within the DMRs of these four genes exhibited a significant negative correlation with gene expression, while no significant correlation was observed for CpGs outside of the DMRs (Fig. S3). In addition, these DMRs also effectively discriminated HCCs from NATs, with AUC values for the four DMRs ranging from 0.72 to 0.93 (Fig. S4).

HCC can be stratified by the genes involved in retinol metabolism

To validate the findings, the methylation and expression of the four retinol metabolism genes in an independent TCGA HCC cohort (365 HCCs, 50 matched normal tissues) were determined. A total of 12 probes could be mapped to the four genes. Except for one probe on the CYP2A6 gene, the methylation status of other 11 probes are nearly consistent with that of WGBS (Fig. 4A). For example, two probes of ADH1A showed hypomethylation in HCC, consistent with prior results (Fig. S5A). Similar results were observed for the probes of the other three genes (Figs. S5B–S5D). However, only six probes were significantly differentially methylated after FDR correction (Table S8), highlighting the limited number of CpGs identified by 450k arrays in resolving methylation patterns.

Figure 4 The expression patterns of retinol metabolism-related genes in HCCs.

(A) Methylation values of 12 probes in TCGA HCC dataset. (B) Expression values of three retinol metabolism-related genes in TCGA HCC dataset. (C) The clustering results of TCGA HCC samples by k-means method. (D) t-SNE plot showing the three subgroups of TCGA HCC samples. (E) Survival curves of three subgroups in TCGA HCC dataset. (F) The clustering results of ICGC LIHC samples by k-means method. (G) t-SNE plot showing the three subgroups of ICGC LIHC samples. (H) Survival curves of three subgroups in ICGC LIHC dataset.

Regarding the TCGA HCC cohort, three of the four retinol metabolism genes had available expression data. All three genes showed significant downregulation in tumors compared to normal tissues (Fig. 4B), validating our findings. Interestingly, substantial heterogeneity was found in the expression of these genes across the HCCs, with some HCCs exhibiting higher and other showing lower levels of expression. The K-means method was employed in the unsupervised clustering analysis, which stratified HCCs into three different subgroups based on retinol gene expression (Fig. 4C). Subgroup 1 (C1, n = 214) displayed the highest expression of the three genes, while subgroup 2 (C2, n = 118) and subgroup 3 (C3, n = 33) showed lower expression levels (Fig. S6A). The UMAP visualization confirmed that the three genes effectively separated HCCs into distinct subtypes (Fig. 4D).

The identified HCC subgroups based on retinol gene expression were associated with distinct survival outcomes. Patients in subgroup C1 demonstrated significantly improved 5-year survival compared to those in subgroups C2 and C3 (Fig. 4E). To validate the findings, an independent liver cancer cohort from ICGC LIHC was analyzed in an identical manner. Remarkable degree of concordance was observed among the cohorts for clustering results (Fig. 4F) and the UMAP visualization (Fig. 4G). Moreover, 5-year survival rate in the ICGC dataset differed significantly between subgroups, but showed a similar trend to the TCGA data (Fig. 4H). In summary, the retinol metabolism genes effectively stratified HCCs into subtypes with prognostic differences, which were validated in two independent large cohorts. These results reveal the potential of retinol gene signatures as biomarkers for more precise 5-year prognosis prediction and treatment guidance in HCC patients.

Clinical features and genomic characteristics of the three subgroups

To characterize differences between HCC subgroups, the average expression of the three retinol genes was calculated as a “retinol score” for each TCGA HCC patient. The distribution of scores varied significantly by clinical feature, with lower scores associated with more advanced, aggressive HCC overall. For instance, patients at stage T1 had higher retinol scores than T2 and T3 (Fig. S6B). Similarly, patients with lymph node metastases had lower scores than those without (Fig. S6C). Significant difference was found among patients with M0, M1 and Mx (Fig. S6D). Retinol scores decreased progressively from AJCC stage I to III and grade G1 to G4 (Figs. S6E–S6F). Patients with HBV or HCV infection did not display significant difference in the retinol scores (Fig. S6G). Notably, younger patients (<50 years) had lower scores (Fig. S6H), possibly reflecting their overexpression among late-stage, high-grade cases in this cohort.

We next examined mutation profiles in the three HCC subgroups (Fig. 5A). The most frequently mutated genes in HCC, including TP53, CTNNB1, TTN, MUC16, and PCL, were common to all groups (Figs. S7A–S7C). There was no significant difference in tumor mutation burden among subgroups (Fig. S7D). However, mutational enrichment analysis revealed subtype-specific mutated genes MDN1, NLGN4X, and COL11A1 were enriched in C1; while PREX2, AXIN1, and CTNNB1 were enriched in C2; and OBSCN, BAP1, and TSC2 were enriched in C3 (Fig. 5B). All subgroups shared the aristolochic acid exposure signature SBS22 (Fig. S8). In summary, although the major HCC mutations were shared, distinct subgroups exhibited a higher prevalence of specific mutated genes, indicating underlying biological differences.

Figure 5 Clinical features and genomic characteristics of the three subgroups.

(A) Mutation profiles of the three subgroups in TCGA HCC dataset. (B) Results of enriched mutations between three subgroups in TCGA HCC dataset. (C) Boxplot showing the proliferation scores between the three subgroups. (D) Boxplot showing the wound-healing scores between the three subgroups. (E) Boxplot showing the IFN-γ response scores between the three subgroups. (F) Boxplot showing the TGF-β response scores between the three subgroups. (G) Boxplot showing the macrophage M0 scores between the three subgroups. (H) Boxplot showing the memory B cells scores between the three subgroups. (I) Boxplot showing Th17 cells scores between the three subgroups. (J) Boxplot showing resting mast cells scores between the three subgroups. P values was estimated by the Kruskal-Wallis test.

We then explored differences in tumor microenvironment (TME) characteristics among HCC subgroups. Cell proliferation and wound healing scores increased from C1 to C3, indicating a gradual increase in tumor cell aggressiveness and angiogenesis in C3 (Figs. 5C–5D). Additionally, C3 showed the lowest IFN-γ but highest TGF-β scores of the three subgroups, indicating poorer immune surveillance (Figs. 5E–5F). Among 24 immune cell types, retinol scores were significantly altered in naive B cells, monocytes, resting NK cells, Th2 cells, M1 macrophages, neutrophils, activated memory CD4 T cells, resting memory CD4 T cells and Tregs (Fig. S9). C3 had the highest scores in M0 macrophage and memory B cells (Figs. 5G–5H) but lowest scores in Th17 cells and resting mast cells (Figs. 5I–5J). Furthermore, there was a gradual increase in the immune score from C1 to C3, with statistically significant differences observed between the three subgroups. However, no significant difference was found in the stromal score (Figs. S10A–S10B). The ESTIMATE score also exhibited an increasing trend but only marginal statistical difference between the three groups could be found (Fig. S10C). Although no statistical difference in the expression level of PD-L1 could be found, the expression level of CTLA4 increased significantly from subgroup C1 to C3 (P < 0.001, Figs. S10D–S10E). Together, these results demonstrate a compromised anti-tumor immune response and a more immunosuppressive TME associated in the C3 subgroup, characterized by low retinol gene expression.

Integration of single-cell RNA sequencing (scRNA seq) data revealed potential functions of retinol metabolism-related genes

To elucidate the biological processes associated with these genes, we integrated scRNA seq data from HCC patients and investigated the expression patterns of the three-retinol metabolism-related genes across various cell types. The scRNA seq dataset, GSE151530, was used in this study. It comprises 49,432 cells from 23 tumor tissues after quality control. These cells were classified into six subgroups, namely T cells, B cells, cancer-associated fibroblasts (CAFs), tumor-associated macrophages (TAMs), tumor-associated endothelial cells (TECs), and epithelial cells, according to the annotation of Gao et al. (Fig. 6A). We calculated the retinol scores for all cells using the expression values of the three genes. Malignant cells have the highest scores and exhibit significant variation, while the retinol scores for other cells were relatively low (almost close to 0) (Fig. 6B). Therefore, we focused on malignant cells and re-clustered them. Malignant cells were clustered into 15 subpopulations (Fig. 6C). Subpopulation 3 exhibited the highest scores, while subpopulations 5, 14, 9, and 12 showed the lowest scores (Fig. 6D). Cell types were annotated using data from the Human Primary Cell Atlas, and the majority of cells in subpopulations 5, 14, 9, and 12 were annotated as hepatocytes (Fig. S11). These findings suggest that the subpopulations of malignant cells exhibit significantly varied retinol scores. We then identified marker genes for the 15 subpopulations and conducted a GO-enrichment analysis. The results showed that marker genes of subpopulations 14, 9, and 12 were predominantly associated with the immune system (Fig. 6E). Moreover, marker genes of subpopulation 5 were enriched in processes related to wound healing, endothelium/vasculature development, and cell migration (Fig. 6E). In addition, the DEGs derived from the comparison between HCC-C1 and HCC-C3 were found to share 91 genes with the marker genes from subpopulation 5. These shared genes were significantly enriched in biological processes such as cell-substrate adhesion, regulation of actin cytoskeleton, and wound healing (Figs. S12A–S12B). In summary, scRNA seq analysis demonstrates retinol metabolism genes are downregulated in specific malignant hepatocyte populations where they may promote proliferation, migration, and angiogenesis while suppressing anti-tumor immunity.

Figure 6 Analysis of retinol metabolism-related genes in scRNA seq data.

(A) UMAP plot showing six cell types in integrated scRNA seq data. (B) Violin plot showing the retinol scores among six cell types. (C) UMAP plot showing the unsupervised clustering of the malignant cells. (D) Violin plot showing the retinol scores among 15 subpopulations of the malignant cells. (E) The significantly enriched GO terms for the marker genes of subpopulations 5, 14, 9 and 12, respectively.

Discussion

This study comprehensively assessed the methylation and gene expression profiles of human HCCs and NATs using WGBS and RNA-seq. The integration of methylation patterns with gene expression enables the identification of a panel of retinol metabolism-related genes whose expression exhibits negative correlations with their methylation levels. The expression profiles of these genes permit the classification of HCC patients into three distinct subgroups, which show significant variations in 5-year survival rates, immune cell infiltration, and tumor microenvironment. In addition, the scRNA seq data analysis identified a subpopulation of cells with low expression of retinol metabolism-related genes. These cells exhibited highly active proliferation and migration behavior, thereby confirming the functional implications of these genes in malignant cells.

Our findings revealed extensive hypo-methylated events in HCCs compared to NATs. This was evidenced by the bimodal distribution of CpG methylation values in HCC, with approximately 10-fold more hypo-DMRs than hyper-DMRs. Moreover, this study found that the majority of (more than 75%) hyper-DMRs were located within the gene body, while only a minority (less than 10%) were located within the regulatory region of genes. This suggests that genome-wide hypomethylation and site-specific hypermethylation are characteristics of HCC, which is consistent with previous studies (Nishiyama & Nakanishi, 2021).

Hypomethylation events contribute to the activation of gene expression (Good et al., 2018) and chromosome instability (Kawano et al., 2014; Sheaffer, Elliott & Kaestner, 2016), which in turn facilitate the development of cancer. This study revealed significant variations in methylation levels across different genomic regions. The regulatory regions (5′ UTR and enhancer) showed the lowest levels in HCC, despite the global hypomethylation characteristics were observed in patients with this disease. This is in line with the observation that a significant proportion of upregulated DEGs are found in HCC, suggesting an increase in gene expression and the activation of oncogenic pathways during the development of HCC. Recent studies have also emphasized the significance of cancer-associated CpG island hypomethylation in driving the overexpression of oncogenic factors.

Among the 17 downregulated hypermethylated genes, enrichment analysis revealed that retinol metabolism was the most significantly enriched pathway. These genes were also significantly enriched in pathways related to intracellular substance metabolism (Fig. 3F), such as the metabolism of olefinic compounds, hormones, terpenoids, alcohol, and so on. It appears that metabolic abnormalities may play a role in the development and progression of HCC. As an important organ of the human body, the liver is involved in the metabolism of major nutrients including glucose, lipids and amino acids (Tenen, Li & Tan, 2021). HCC has been reported to exhibit various metabolic characteristic changes, such as increased aerobic glycolysis, enhanced de novo lipogenesis, glutamine depletion, imbalance in oxidative metabolism etc. (Satriano et al., 2019). The findings of this study hold significant value in the development of innovative anti-metabolic drugs. Four of the 32 genes involved in retinol metabolism were identified among the 17 genes, representing the highest percentage (23.53%). We therefore focused on the four representative genes among the hyper-downregulated genes. It should be noted, however, that the remaining 13 genes warrant further investigation in future studies.

Correlation analysis revealed a negative correlation between gene expression and the methylation levels of CpGs that located in gene regulatory regions. These findings suggested that methylation events may play a pivotal role in inhibiting expression of genes involved in retinol metabolism. In the TCGA HCC dataset, the methylation profiles of probes associated with the four genes demonstrated a concordance with the results of WGBS. However, no significant negative correlations were observed between the methylation values and gene expression. Further investigations showed that these probes were not located in DMRs identified in this study, which may account for this discrepancy. Besides, only a limited number of probes (12 probes) could be mapped to the four genes, which is related to the fact that the CpG sites detected by the Infinium Human Methylation 450k BeadChip (4.5 × 105 CpG sites) are less than 2% of the genome-wide (2.8 × 107 CpG sites), suggesting the limitations of methylation array technique in certain critical genomic regions.

This study also found that HCC patients could be stratified into three subgroups, based on the expression levels of four retinol metabolism-related genes. HCC patients with low expression levels of the four genes demonstrated a significantly poorer prognosis in 5 years, as well as increased proliferation and wound healing activity. These findings suggest that the abnormal expression of these genes may influence the prognosis of HCC by affecting cell proliferation and expansion. Moreover, the HCC subgroup with the worst 5-year survival rate (HCC-C3) exhibited more immunosuppressive features than the subgroup with the best prognosis (HCC-C1). This suggests that the prognostic associations of these subgroups may also involve immune system remodeling. Interestingly, the scRNA seq data revealed minimal expression of these genes in immune cells and elevated expression in malignant cells. These findings suggest potential molecular mechanisms that link dysregulated expression of retinol metabolism-related genes to the development and progression of HCC.

Several studies have demonstrated that the abnormal expression of genes involved in retinol metabolism can be used as diagnostic and prognostic markers for HCC (Zhao et al., 2022; Awan et al., 2015; Wang et al., 2018). In addition, it has been reported that serum retinol levels can serve as a diagnostic and prognostic marker for HCC, with the low expression of retinol correlated with decreased survival time (Han et al., 2020). In this study, we observed hypermethylation in the regulatory regions of these genes, with significant negative correlations between methylation and expression. This suggests that methylation may play a key role in driving altered gene expression. Meanwhile, we found that these DMRs have the ability to discriminate HCCs from NATs, thus emphasizing their potential for diagnostic purposes.

ScRNA seq data analysis revealed that these four genes were expressed in malignant cells with high variation. In contrast, they were rarely expressed in other cell types, including T cells, B cells, CAFs, TAMs, TECs and epithelial cells. Further investigations showed the lowest expression of the four genes in subpopulations of malignant cells that exhibited activated biological processes of wound healing, endothelium/vasculature development, and cell migration. A recent study has reported that the overexpression of CYP2C8 inhibits the proliferation, clonality, migration, invasion, and cell cycle of HCC cells through the PI3K/Akt/p27Kip1 axis (Zhou et al., 2021). Meanwhile, in vitro experiments also revealed that a synthetic retinoid, ACR, functions as an agonist for RARs and RXRs, inducing apoptosis and inhibiting cell proliferation in human HCC cell lines (Shirakami, Sakai & Shimizu, 2015). Interestingly, our findings also suggest that these genes may act as favorable prognostic factors by inhibiting biological processes such as cell proliferation and migration.

The current study also has certain limitations. First, our analysis was limited to the four hypermethylated genes associated with retinol metabolism. It is possible that other identified genes may also play an important role in the development and progression of HCC, and further research in this area is warranted. Second, while our findings showed that the four hypermethylated genes could distinguish HCCs from NATs, however, further evidence is required to substantiate their diagnostic value in clinical practice using a larger cohort. Third, gene expression is regulated by a complex process. In addition to abnormal epigenetic modifications, other factors such as miRNAs and copy number variations have also been shown to influence gene expression. A significant negative correlation was observed between the methylation of DMRs in the four genes and their expression. Nevertheless, there was no direct evidence to suggest that the observed downregulated of expression was solely due to hypermethylation. Further molecular and cellular experiments are required to investigate these relationships in detail.

Conclusions

We observed widespread demethylation events in HCC accompanied by the activation of gene expression. Four retinol metabolism-related genes were able to categorize HCC patients into three subgroups with different prognoses and immune microenvironments. The integration of scRNA seq data enabled the identification of a cell subset with low retinol scores and active wound healing and proliferation. The present study provides a new, multidimensional perspective on the development and progression of HCC. Additionally, we identified aberrant methylation and expression candidates that may potentially serve as methylation markers, therapeutic targets, and prognostic markers. However, further research is warranted to explore the functions of other identified genes and to evaluate the diagnostic and prognostic performance of these biomarkers in larger cohorts.

Supplemental Information

Figure S1 Distribution of methylation levels of all CpG sites between the three WGBS datasets

(A) The comparison of PRJNA984754 vs. GSE70090. (B) The comparison of PRJNA984754 vs. PRJNA762641. (C) The comparison of GSE70090 vs. PRJNA762641. Y-axis is the density, and x-axis is the difference in methylation values (deltaM) of CpGs between two datasets, ranging from −100 to 100 (percentile). When two datasets are identical, deltaM should be 0 (corresponding to one peak at position 0).

Figure S2 Box-plot shows the overall methylation level of 12 HCC and matched NAT samples

One sample (16A) was excluded because of low sequencing depth.

Figure S3 The correlation of CpG methylation with gene expression for the four genes related to retinol metabolism

The upper part of each panel shows the DMRs, exons/UTRs, and CpGs within the gene. Y-axis indicates the methylation values of CpGs between HCCs and NATs. The lower part of each panel shows the correlation of CpG methylation with gene expression. Red stars represent significantly positive correlations and blue stars represent significantly negative correlations.

Figure S4 ROC curves demonstrate the performance of the methylation values of four genes’ DMRs to discriminate HCCs from NATs

The point in each curve indicates the optimal cut-off value, and the optimal specificity and sensitivity.

Figure S5 Methylation values of the four genes associated 450K probes in TCGA HCC dataset

The orange areas indicate the CpGs’ methylation values determined by WGBS data and the purple rectangles indicate the location of DMRs. Each point represents a probe of 450k array. The Y-axis indicates the methylation difference between HCC and NATs. The x-axis indicates the genomic coordinates of each gene.

Figure S6 The association of retinol scores with clinical characteristics between three HCC subgroups

The retinol scores were calculated using the average expression values of the four genes.

Figure S7 Mutation patterns of the three subgroups in the TCGA HCC dataset

Figure S8 Mutation signatures of the three subgroups

Figure S9 Immune cell scores of the three subgroups

Figure S10 Immune scores, stromal scores, ESTIMATE scores and the expression levels of PD-L1 and CTLA4 of the three subgroups

Figure S11 Cell type annotations for the subpopulations of malignant cells

Figure S12 Comparison of DEGs between TCGA HCC-C1 (highest retinol scores) and HCC-C3 (lowest retinol scores) with marker genes of subpopulation 5

Table S1 The clinical characteristics of 12 HCC patients

Table S2 Sample information for the PRJNA762641 dataset

Table S3 Sample information for the GSE70090 dataset

Table S4 Details of all datasets used in the study

Table S5 The quality of WGBS data for 24 samples

Table S6 The top 100 DMRs with the most significant differences between HCCs and NATs

Table S7 Information on the 17 hyper-downregulated genes

Table S8 Methylation values of the 12 probes associated with the four genes in TCGA HCC dataset

We are grateful to Dr. Xu at the Zhongnan Hospital of Wuhan University for his assistance.

Additional Information and Declarations

Competing Interests

Author Contributions

Human Ethics

Data Availability

Kangkang Wan, Lianglu Zhang, Lanlan Dong, Dihan Zhou and Wei Zhang are employed by Wuhan Ammunition Life-tech Company, Ltd.

Yanteng Zhao conceived and designed the experiments, analyzed the data, prepared figures and/or tables, authored or reviewed drafts of the article, and approved the final draft.

Kangkang Wan conceived and designed the experiments, analyzed the data, prepared figures and/or tables, authored or reviewed drafts of the article, and approved the final draft.

Jing Wang performed the experiments, analyzed the data, prepared figures and/or tables, and approved the final draft.

Shuya Wang performed the experiments, analyzed the data, authored or reviewed drafts of the article, and approved the final draft.

Yanli Chang performed the experiments, analyzed the data, authored or reviewed drafts of the article, and approved the final draft.

Zhuanyun Du performed the experiments, analyzed the data, authored or reviewed drafts of the article, and approved the final draft.

Lianglu Zhang conceived and designed the experiments, authored or reviewed drafts of the article, and approved the final draft.

Lanlan Dong conceived and designed the experiments, authored or reviewed drafts of the article, and approved the final draft.

Dihan Zhou performed the experiments, prepared figures and/or tables, and approved the final draft.

Wei Zhang performed the experiments, prepared figures and/or tables, and approved the final draft.

Shaochi Wang conceived and designed the experiments, authored or reviewed drafts of the article, and approved the final draft.

Qiankun Yang conceived and designed the experiments, authored or reviewed drafts of the article, and approved the final draft.

The following information was supplied relating to ethical approvals (i.e., approving body and any reference numbers):

This study was approved by the Ethics Committee of the First Affiliated Hospital of Zhengzhou University with approval number 2022-KY-0631-002.

The following information was supplied regarding data availability:

The raw data of 24 samples (12 HCC and 12 NATs) are available at the Sequence Read Archive database: PRJNA98475.

The methylation data and associated R code used in this study are available on GitHub and Zenondo: https://github.com/amsinfor/HCC-WGBS-analysis.

amsinfor. 2024. amsinfor/HCC-WGBS-analysis: Code (Code). Zenodo. https://doi.org/10.5281/zenodo.12047383.

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
