# Peer review of "DNA methylation and gene expression profiling reveal potential association of retinol metabolism related genes with hepatocellular carcinoma development"

_PeerJ, doi:10.7717/peerj.17916_

## Round 0.1 · original submission · Major Revisions

Please carefully consider all point raised by the reviewers.

**Language Note:** PeerJ staff have identified that the English language needs to be improved. When you prepare your next revision, please either (i) have a colleague who is proficient in English and familiar with the subject matter review your manuscript, or (ii) contact a professional editing service to review your manuscript. PeerJ can provide language editing services - you can contact us at [email protected] for pricing (be sure to provide your manuscript number and title). – PeerJ Staff

Reviewer 1 ·

Basic reporting

English is clear, I only suggest to modify this sentence at line 123:
The GSE151530 (Ma et al., 2021) single-cell RNA-seq dataset contains 46 samples ....


I think Figure 5 should be modified: please use the same colours to identify the clusters in all panels (eg C1 always in red, C2 in blue, C3 in red)

Experimental design

The overall work is interesting, however I would like to present some points that should be improved:

1 The application of data fusion/integration methods could be useful to statistically validate the relationship between gene expression and methylation (with methods such as MethylMix). The application of multiple genomic domains integration could be useful to check any overlap between defined clusters and the ones defined by data integration (as RNAseq, methylation and SNV data), this could be potentially tested with the Movics R package (https://github.com/xlucpu/MOVICS) or other methods (as the ones tested in this work: Evaluation and comparison of multi-omics data integration methods for cancer subtyping).

Minor suggestions

2 Consider using package like singscore to classify samples or evaluate gene signatures, in my experience it is generally better compared to a sum/average over selected gene counts or k-means.

Validity of the findings

no comment

Reviewer 2 ·

Basic reporting

The manuscript investigated DNA methylation and gene expression profiling based on bioinformatics and wet-lab experiments. 4 retinol metabolism-related genes were identified as indicators for prognoses and immune microenvironments for hepatocellular carcinoma. The findings are a supplement to existing literature to some extent.

The language is intelligible and accurate in reporting and stating idea.

The Introduction section unfolds logically with a funnel structure. Authors mentioned the relationship between methylation and the development of hepatocellular cell carcinoma, then indicated research gap and the aims of this manuscript. Authors reviewed relevant literatures properly. The background was demonstrated sufficiently.

However, in Line 86 -88, authors mentioned that a few attempts have obtained valuable findings and mentioned two literatures and said that these findings are insufficient. I recommend the authors could introduce the two literatures briefly to help readers to know why these findings are insufficient.

Experimental design

The investigation was rigorously conducted by using bioinformatic methods and wet-lab experiments. Methods were described with relatively adequate information, which helps other investigators to replicate.

From my perspective, there are still some issues need to be solved:

1) The authors selected the dataset to screen the differential expression profiling between hepatocellular carcinoma and adjacent normal tissues, there was no distinction between virus-associated hepatocellular carcinoma and non-virus-associated hepatocellular carcinoma. However, based on current knowledge, differences exist between virus-associated and non-virus-associated hepatocellular carcinoma in terms of pathogenesis, gene expressions, immune microenvironment, and prognosis, which may lead to confounding and bias in the results.

2) Line 134-135: “The retinol score for each cell was characterized by the average expression of three genes - CYP2C8, CYP2A6, and ADH1A”. What is your basis for adopting this method? Any relevant literature?

3) Could you add etiology information in Supplementary table 2 and Supplementary table 3?

4) The manuscript revealed that the genes may have an impact on immune system. Thus, if possible, I recommend the authors investigate the difference of immune microenvironment of the 3 subgroups, such as stromal score, immune score, and ESTIMATE score, as well as PD-L1 and CTLA4 expression level, etc.

Validity of the findings

Although dozens of articles focused on retinol metabolism related genes and hepatocellular carcinoma, the manuscript integrated information of methylation and the above genes, it is still a contribution to the field.

For the results section and discussion section, recommendations are listed below:

1) Line 272-285: I recommend the authors only remain the findings of this study in the Result section, and put the findings from previous literatures into the Discussion section if possible.

2) Line 313: Could you please list the 17 hyper-downregulated genes? Maybe you can list in the text or in a supplementary table.

3) The author mentioned the 4 of 32 retinol metabolism pathway genes are (hyper) down-regulated hypermethylated genes (Line 445 – 448). How about the other 28 retinol metabolism pathway genes? Are they hyper-upregulated, hyper-downregulated, hypo-upregulated, or hypo-downregulated? Maybe you can list in a supplementary table.

4) Why chose the 4 retinol metabolism pathway genes rather than the other 11 hyper-downregulated genes for further study? The authors stated reasons why chose the 4 genes in the Discussion section, but I still recommend authors discuss this issue pro and con.

Additional comments

None

Reviewer 3 ·

Basic reporting

no comment

Experimental design

no comment

Validity of the findings

no comment

Additional comments

In this manuscript, the authors analyzed multi-omics data (DNA methylation and gene expression) from various projects to study the molecular mechanisms of the retinol metabolism-related genes underlying hepatocellular carcinoma to imply future diagnosis, prognosis and treatment. Especially, the authors also integrated scRNA-seq technology and evaluated both diagnostic and prognostic effect of the same biomarkers. Overall, the manuscript is well written and organized, with clear object and sufficient results in data analysis. However, several modifications could to be done on clarifying the methods and strengthen the conclusion.

Here are my major comments:

1. The authors integrated 3 datasets from different sources into one for differential methylation analysis. Although the methylation patterns are consistent across 3 datasets as shown in Figure 1, I'm wondering if any within- and between- data normalization methods were used as this is a typical pre-processing step. In addition, are all data from 450K microarray sequencing? If not, would such integration exclude many probes?

2. Line 163-172, you calculated the delta value from the average methylation value to indicate the DMRs. Which value was used in the calculation, M or beta? People usually used delta beta value, where the cutoff should not be as described in the paper (>=10 and <=-15). To me, this is not a sound cutoff method and please explain more details and provide evidence.

3. For the DE analysis, the authors only analyzed one of the datasets from their chosen project.
a. The sample size is substantially less than in methylation analysis., do you expect any bias or loss of power? People usually keep samples without missing data, i.e. same sample with both methylation and gene expression data. I suggest to put more evidence to support this and also discuss it in the discussion section.

b. Line 201, rank-sum test was used for DEG identification. Does rank-sum test here refer to Wilcoxon rank sum test? If so, this is not the appropriate statistical method for here, since it is paired sample from the same person, which means the HCC and NAT are dependent samples. Under this case, Wilcoxon signed-rank test should be used. Please clarify.

4. For all of you statistical analysis, are there any covariates, e.g., age, sex being adjusted? Also, all your own HCC samples are from male (table S1) whereas the other 2 public datasets have both male and female samples, is there a bias on sample selection?

5. In the results section, the authors mentioned their integrated analysis of DNA methylation and gene expression are from the same HCC patient (line 292-293), which is conflict with the description of the methods before. Please clarify more details here.

6. The authors studied the progression of identified 4 retinol genes and get a Kaplan-Meier curve (line 339-348 and Figure 4). Can you also provide the ROC curve with AUC and concordance index as the measure of prediction accuracy in addition to the Kaplan-Meier curve? I also suggest to extend the analysis by including some comparisons with existing biomarkers etc. to make your conclusion more convinced.

7. Line 329, the authors stated the limitations of 450k array maybe the cause of small number of identified DMC from TCGA data, but without provided further explanations. This is not straightforward to me since sample size and many other factors can also be the potential reason. Do you expect advanced sequencing like 850k or 900k to improve this? But with more probes being sequenced, there is more stringent FDR control threshold so I'm wondering why the authors made this statement. I suggest to refine the sentences and include more details in the discussion section.

---

## Round 0.2 · Minor Revisions

Please address all issues raised.

Reviewer 1 ·

Basic reporting

no comment

Experimental design

no comment

Validity of the findings

no comment

Additional comments

no comment

Reviewer 2 ·

Basic reporting

The authors have provided very conscientious responses and revisions to the comments. The responses have answered the questions I put forward to a large extent, and the manuscript has been significantly enhanced.

However, I still have a suggestion: the authors provided a detailed analysis and response to the following recommendation in the response letter, and obtained meaningful and positive results, Nevertheless, it seems that the relevant content has not been incorporated into the manuscript. I suggest the authors present the content in the section of MATERIALS & METHODS and RESULTS.

Previous recommendation: “The manuscript revealed that the genes may have an impact on immune system. Thus, if possible, I recommend the authors investigate the difference of immune microenvironment of the 3 subgroups, such as stromal score, immune score, and ESTIMATE score, as well as PD-L1 and CTLA4 expression level, etc.”

No other comments.

Experimental design

N/A

Validity of the findings

N/A

Additional comments

N/A

Reviewer 3 ·

Basic reporting

no comment

Experimental design

no comment

Validity of the findings

no comment

Additional comments

The authors have addressed most of my comments. I only have one minor follow-up concern:

I appreciated the authors looked into the AUC values on survival prediction for the 4 retinol genes. However, the AUC value around 0.6 does not indicate a very high prediction accuracy. Along with Figure 4E, subgroup C1 has a very poor survival rate after 5-6 years, but in the revised manuscript line 355-356, it said C1 had significantly improved overall survival compared to C2 and C3. Based on the independent validation data showed in Figure 4H, samples from this dataset does not have survival information after 6 year where no comparison can be made with the TCGA data. I think identification of the 4 genes in current manuscript is important, but I'd suggest not to over-state the conclusion.

Also, although may out of scoop of this manuscript, it's interesting to look into the samples from TCGA data whose survival is longer than 6 years, and understand what happened or consider the cause of poor classification there in the future. I'm curious about how does the authors discuss this, and will it be linked with high heterogeneity in HCC and indicate potential personalized therapies?

---

## Round 0.3 · accepted · Accept

The authors have adequately addressed all issues raised. The manuscript is now ready for publication.